# Generalization bounds for mixing processes via delayed online-to-PAC conversions

## Abstract

We study the generalization error of statistical learning algorithms in a non-*i.i.d.* setting, where the training data is sampled from a stationary mixing process. We develop an analytic framework for this scenario based on a reduction to online learning with delayed feedback. In particular, we show that the existence of an online learning algorithm with bounded regret (against a fixed statistical learning algorithm in a specially constructed game of online learning with delayed feedback) implies low generalization error of said statistical learning method even if the data sequence is sampled from a mixing time series. The rates demonstrate a trade-off between the amount of delay in the online learning game and the degree of dependence between consecutive data points, with near-optimal rates recovered in a number of well-studied settings when the delay is tuned appropriately as a function of the mixing time of the process.

## 1 Introduction

In machine learning, generalization denotes the ability of a model to infer patterns from a dataset of training examples and apply them to analyze previously unseen data (Shalev-Shwartz and Ben-David, 2014). The gap in accuracy between the model's predictions on new data and those on the training set is usually referred to as *generalization error*. Providing upper bounds on this quantity is a central goal in statistical learning theory. Classically, bounds based on notions of complexity (*e.g.*, VC dimension and Rademacher complexity) for the model's hypothesis space were used to provide uniform worst-case guarantees (see Bousquet et al., 2004; Vapnik, 2013; Shalev-Shwartz and Ben-David, 2014). However, results of this kind are often too loose to be applied to the most common machine learning over-parameterised models, such as deep neural networks (Zhang et al., 2021). As a consequence, several approaches have been proposed to obtain algorithm-dependent generalization bounds, which can adapt to the problem and be much tighter in practice than their uniform counterparts. Often, the underlying idea is that if the algorithm's output does not have a too strong dependence on the specific input dataset used for the training, then the model should not be prone to overfitting, and so generalize well. Examples of results that build onto these ideas are stability bounds, information-theoretic bounds, and PAC-Bayesian bounds (see, *e.g.*, Bousquet and Elisseeff, 2002; Russo and Zou, 2020; Hellström et al., 2023; Alquier, 2024).

Most results in the literature focus on the *i.i.d.* setting, where the training dataset is made of independent draws from some underlying data distribution. However, for several applications, this assumption is far from realistic. For instance, it excludes the case where observations received by the learner have some inherent temporal dependence, as it is the case for stock prices, daily energy consumption, or sensor data from physical environments (Ariyo et al., 2014; Takeda et al., 2016). This calls for the development of theory for addressing non-*i.i.d.* data. A common approach in the extant literature is to consider a class of non-*i.i.d.* data-generating processes usually referred to as stationary $\beta$-mixing

or $\varphi$-mixing processes. This assumption, together with a "blocking" trick introduced by Yu (1994), has led to a few results in the literature: Meir (2000), Mohri and Rostamizadeh (2008), Shalizi and Kontorovich (2013), and Wolfer and Kontorovich (2019) provided uniform worst-case generalization bounds, Steinwart and Christmann (2009) and Agarwal and Duchi (2012) discussed excess risk bound (comparing the algorithm's output with the best possible hypothesis), while Mohri and Rostamizadeh (2010) gave bounds based on a stability analysis (in the sense of Bousquet and Elisseeff, 2002).

Here, we propose propose results for the non-*i.i.d.* setting in the form of PAC-Bayesian bounds (Guedj, 2019; Alquier, 2024): high probability upper bounds on the expected generalization error of randomized learning algorithms. We achieve this by combining the "blocking" argument by Yu (1994) to manage the concentration of sums of correlated random variables, with the recent online-to-PAC conversion technique recently proposed by Lugosi and Neu (2023). Using their framework we show a new way to obtain generalization bounds for stationary dependent processes that satisfy a certain "short-memory" property (intuitively meaning that data points that are closer in time are more heavily dependent on each other). Our assumption slightly differs from $\beta$-mixing in the sense that we only need it to hold for a specific class of bounded loss functions. Among other results, this allows us to prove PAC-Bayesian generalization bounds for mixing processes. This complements previous work on such bounds that have only considered mild relaxations of the *i.i.d.* condition such as assuming that the data has a martingale structure (see, *e.g.*, Seldin et al., 2012; Chugg et al., 2023; Haddouche and Guedj, 2023). Notable exceptions are the works of Alquier and Wintenberger (2012), Alquier et al. (2013), and Eringis et al. (2022, 2024), who provided generalization bounds for a sequential prediction setting where both the data-generating process and the hypothesis class used for prediction are stable dynamical systems. Their results are proved under some very specific conditions on these systems, and their guarantees involve unspecified problem-dependent constants that may be large. In contrast, our bounds hold under general, simple-to-verify conditions and feature explicit constants.

The rest of the paper is organized as follows. In Section 2 we properly define the generalization error of a statistical learning algorithm for both *i.i.d.* and non-*i.i.d.* cases, and state our main assumption on the data dependence. Our main contribution lies in Section 3, where after recalling the results for the *i.i.d.* setting we show how to adapt this to stationary mixing processes. In Section 4 we provide concrete results of the bounds we can obtain through the online-to-PAC conversion. Finally in Section 5 we extend our results to the setting where the hypothesis class itself may consist of dynamical systems.

**Notation.** For a distribution over hypotheses $P \in \Delta_W$ and bounded function $f : \mathcal{W} \to \mathbb{R}$ we write $\langle P, f \rangle$ to refer to the expectation of $\mathbb{E}_{W \sim P}[f(W)]$. We denote $\mathcal{D}_{KL}(P||Q) = \mathbb{E}_{X \sim P}\left[\ln\left(\frac{P(X)}{Q(X)}\right)\right]$ to refer to the Kullback-Leibler divergence. We use $||.||$ to denote a norm on the Banach space $\mathcal{Q}$ of the finite signed measures, and $||.||_*$ the corresponding dual norm on the dual space $\mathcal{Q}^*$ of measurable functions $f$ on $\mathcal{W}$ such that $||f||_* = \sup_{Q \in \mathcal{Q}:||Q|| \leq 1} \langle Q, f \rangle$.

## 2 Preliminaries

The classical statistical learning framework usually considers a dataset $S_n = (Z_1, ..., Z_n)$, made of $n$ *i.i.d.* elements drawn from a distribution $\mu$ over a measurable instance space $\mathcal{Z}$. Often, one can think of each $Z_i$ as a feature-label pair $(X_i, Y_i)$. Furthermore, we are given a measurable class $\mathcal{W}$ of hypotheses and a loss function $\ell : \mathcal{W} \times \mathcal{Z} \to \mathbb{R}_+$, with $\ell(w, z)$ measuring the quality of the hypothesis $w \in \mathcal{W}$ on the data instance $z \in \mathcal{Z}$. For any given hypothesis $w \in \mathcal{W}$, two key objects of interest are the *training error* $\widehat{\mathcal{L}}(w, S_n) = \frac{1}{n}\sum_{i=1}^{n} \ell(w, Z_i)$ and the *test error* $\mathcal{L}(w) = \mathbb{E}_{Z' \sim \mu}[\ell(w, Z')]$, where the random element $Z'$ has the same distribution as $Z_i$ and is independent of $S_n$.

A learning algorithm $\mathcal{A} : \mathcal{Z}^n \to \mathcal{W}$ maps the training sample to an hypothesis in $\mathcal{W}$. More generally, we will focus on randomized learning algorithms, returning a probability distribution $P_{W_n|S_n} \in \Delta_{\mathcal{W}}$ over $\mathcal{W}$, conditionally on $S_n$ (deterministic algorithms can be recovered as special cases, whose the outputs are Dirac distributions). The ultimate goal of the learner is to minimize the test error. Yet, this quantity cannot be computed without knowledge of the data generating distribution $\mu$. In practice, one typically relies on the training error in order to gauge the quality of the algorithm. For an algorithm $\mathcal{A} : S_n \mapsto P_{W_n|S_n}$, we define the *generalization error* as the expected gap between training and test error:

$$\text{Gen}(\mathcal{A}, S_n) = \mathbb{E}\left[\mathcal{L}(W_n) - \widehat{\mathcal{L}}(W_n, S_n)\Big| S_n\right].$$

The expectation in the above expression integrates over the randomness in the output of the algorithm $W_n \sim P_{W_n|S_n}$, conditionally on the sample $S_n$. We remark that the test error is *not* equal to the mean of the training error, due to the dependence of $W_n$ on the training data.

We extend the previous setting by considering the case where the data have an intrinsic temporally ordered structure, and come in the form of a stationary process $(Z_t)_{t \in \mathbb{N}^*} \sim \nu$. Formally, we assume that the joint marginal distribution of any block $(Z_t, Z_{t-1}, \ldots, Z_{t-i})$ is the same as the distribution of $(Z_{t+j}, Z_{t+j-1}, \ldots, Z_{t+j-i})$ for any $t$, $i$ and $j$, but the data points are not necessarily independent of each other. In particular, the marginal distribution of $Z_t$ is constant and is denoted by $\mu$. Thus, it is natural to continue to use the definition of the test loss and generalization error given above, although with the understanding that $\mu$ now refers to the marginal distribution of an independent copy of $Z_1$, a sample point from a stationary non-*i.i.d.* process. We remark here that other notions of the test loss may also be considered, and the framework that we propose can be extended to most natural definitions with little work (but potentially large notational overhead). In Section 5, we provide such an extension for a more general setting where the hypotheses themselves are allowed to have memory and the process may not be as strongly stationary as our assumption above requires.

In order to obtain generalization results we need to have some control on how strong the dependencies between different datapoints are allowed to be. To this regard, we consider the following assumption.

**Assumption 1.** *There exists a non-increasing sequence $(\phi_d)_{d \in \mathbb{N}^*}$ of non-negative real numbers such that, for all $w \in \mathcal{W}$ and all $t \in \mathbb{N}^*$:*

$$\mathbb{E}\left[\mathcal{L}(w) - \ell(w, Z_t) \Big| \mathcal{F}_{t-d}\right] \leq \phi_d \,,$$

*where $\mathcal{L}(w) = \mathbb{E}_{Z' \sim \mu}[\ell(w, Z')]$, with $Z'$ being independent on the process $(Z_t)_{t \in \mathbb{N}^*}$ and having as distribution the stationary marginal $\mu$ of the $Z_t$.*

The intuition behind this assumption is that the loss associated with the observations $Z_t$ becomes almost independent of the past after $d$ steps, enabling us to treat each sequence of the form $(Z_t, Z_{t+d}, \ldots, Z_{t+(n-t)d})$ as an approximately *i.i.d.* sequence. Note that this assumption differs from the usual $\beta$-mixing assumption which requires the distribution of $Z_t|\mathcal{F}_{t-d}$ to be close to the marginal distribution $\mu$ for all $t$, in terms of total variation distance. Our assumption is somewhat weaker in the sense that it only requires the expected losses under these distributions to be close, and only a one-sided inequality is required. It is easy to verify that our assumption is satisfied if the process is $\beta$-mixing in the usual sense and the losses are bounded in $[0, 1]$.

# 3 Proving generalization bounds via online learning

Online learning focuses on algorithms that aim to improve performance incrementally as new information becomes available, often without any underlying assumption on how data are generated. The online learner's performance is typically measured leveraging the idea of regret. This involves introducing a cost function for the problem and defining the regret as the difference between the cumulative cost of the online learner and that of a fixed comparator. We refer to the monographs Cesa-Bianchi and Lugosi, 2006 and Orabona, 2019 for comprehensive overviews on online learning and regret analysis. Recently, Lugosi and Neu (2023) established a connection between upper bounds on the regret and generalization bounds, showing that the existence of a strategy with a bounded regret in a specially designed online game translates into a generalization bound, via a technique dubbed *online-to-PAC conversion*. Their focus is on the *i.i.d.* setting, where the training dataset is made of independent draws. Here, we show that this framework can naturally be extended beyond the *i.i.d.* assumption.

In what follows, we briefly review the setup of Lugosi and Neu (2023) in Section 3.1 and then describe our new extension of their model to the non-*i.i.d.* case in Section 3.2. In particular, we prove a high-probability bound for the generalization error of any statistical learning algorithm learnt with a stationary mixing process verifying Assumption 1.

## 3.1 Online-to-PAC conversions for *i.i.d.* data

Lugosi and Neu (2023) have recently established a framework to obtain generalization bounds via a reduction to online learning. Their technique allows to recover several classic PAC-Bayesian

results, and provide a range of generalizations thereof. The main idea of Lugosi and Neu (2023) is to introduce an online learning game called the *generalization game*, where the following steps are repeated for a sequence of rounds $t = 1, 2, \ldots, n$:

- the online learner picks a distribution $P_t \in \Delta_{\mathcal{W}}$;
- the adversary selects the cost function $c_t : w \mapsto \ell(w, Z_t) - \mathcal{L}(w)$;
- the online learner incurs the cost $\langle P_t, c_t \rangle = \mathbb{E}_{W \sim P_t}[c_t(W)]$;
- $Z_t$ is revealed to the learner.

The learner can adopt any strategy to pick $P_t$, but they can only rely on past knowledge to make their prediction. Explicitly, if $\mathcal{F}_t$ denotes the sigma-algebra generated by $Z_1, ..., Z_t$, then $P_t$ has to be $\mathcal{F}_{t-1}$-measurable. We also emphasize that in this setup the online learner is allowed to know the loss function $\ell$ and the distribution $\mu$ of the data points $Z_t$, and therefore by revealing the value of $Z_t$, the online learner may compute the entire cost function $c_t$.

We define the *regret* of the online learner against the possibly data-dependent *comparator* $P^* \in \Delta_{\mathcal{W}}$ as $\mathrm{Regret}(P^*) = \sum_{t=1}^n \langle P_t - P^*, c_t \rangle$. Now, denote as $P_{W_n|S_n}$ the distribution produced by the supervised learning algorithm. With this notation, the generalization error can be written as $\mathrm{Gen}(\mathcal{A}, S_n) = -\frac{1}{n} \sum_{t=1}^n \langle P_{W_n|S_n}, c_t \rangle$. By adding and subtracting the quantity $M_n = -\frac{1}{n} \sum_{t=1}^n \langle P_t, c_t \rangle$ we get the following decomposition.

**Theorem 1** (Theorem 1 in Lugosi and Neu, 2023; see appendix A.1). *With the notation introduced above,*

$$\mathrm{Gen}(\mathcal{A}, S_n) = \frac{\mathrm{Regret}_n(P_{W_n|S_n})}{n} + M_n . \tag{1}$$

The first of these terms correspond to the *regret* of the online learner against a fixed *comparator strategy* that picks $P_{W_n|S_n}$ at each step. The second term is a martingale and can be bounded in high probability with standard concentration tools. Indeed, since $P_t$ is chosen before $Z_t$ is revealed, one can easily check that $\mathbb{E}[\langle P_t, c_t \rangle | \mathcal{F}_{t-1}] = 0$. Thus, to prove a bound on the generalization error of the statistical learning algorithm, it is enough to find an online learning algorithm with bounded regret against $P_{W_n|S_n}$ in the generalization game.

As a concrete application of the above, the following generalization bound is obtained when picking the classic exponential weighted average (EWA) algorithm (Vovk, 1990; Littlestone and Warmuth, 1994; Freund and Schapire, 1997) as online strategy, and plugging its regret bound into (1).

**Theorem 2** (Corollary 6 in Lugosi and Neu, 2023). *Suppose that $\ell(w, z) \in [0, 1]$ for all $w, z$. Then, for any $P_1 \in \Delta_{\mathcal{W}}$ and $\eta > 0$, with probability at least $1 - \delta$ on the draw of $S_n$, uniformly on every learning algorithm $\mathcal{A} : S_n \mapsto P_{W_n|S_n}$, we have*

$$\mathrm{Gen}(\mathcal{A}, S_n) \leq \frac{\mathcal{D}_{KL}(P_{W_n|S_n} || P_1)}{\eta n} + \frac{\eta}{2} + \sqrt{\frac{2 \log\left(\frac{1}{\delta}\right)}{n}} .$$

*Proof.* We can bound each term of (1) separately. A data-dependent bound for the regret term is obtained via a direct application of the regret analysis of EWA which brings the term $\frac{\mathcal{D}_{KL}(P_{W_n|S_n} || P_1)}{\eta n} + \frac{\eta}{2}$ (see Appendix B.1). The term $\sqrt{\frac{2 \log\left(\frac{1}{\delta}\right)}{n}}$ results from bounding the martingale $M_n$ via an application of Hoeffding–Azuma inequality. $\square$

Note that the first term in the above bound is data-dependent due to the presence of $P_{W_n|S_n}$, and thus optimizing it requires a data-dependent choice of $\eta$, which is not allowed by Theorem 2. However, via a union bound argument it is possible to get a bound in the form

$$\mathrm{Gen}(\mathcal{A}, S_n) = \mathcal{O}\left( \sqrt{\frac{\mathcal{D}_{KL}(P_{W_n|S_n} || P_1)}{n}} + \sqrt{\frac{1}{n} \log\left(\frac{\log n}{\delta}\right)} \right),$$

For the details, we refer to the proof of Corollary 5 of Lugosi and Neu (2023), which recovers a classical PAC-Bayes bound of McAllester (1998).

## 3.2 Online-to-PAC conversions for non-*i.i.d.* data

In what follows, we will drop the *i.i.d.* assumption for the data, and instead consider non-*i.i.d.* sequences satisfying Assumption 1. For this setting we define the following variant of the generalization game.

**Definition 1** (Generalization game with delay). *The generalization game with delay $d \in \mathbb{N}^*$ is an online learning game where the following steps are repeated for a sequence of rounds $t = 1, ..., n$:*

- *the online learner picks a distribution $P_t \in \Delta_{\mathcal{W}}$;*

- *the adversary selects the cost function $c_t : w \mapsto \ell(w, Z_t) - \mathcal{L}(w)$;*

- *the online learner incurs the cost $\langle P_t, c_t \rangle = \mathbb{E}_{W \sim P_t}[c_t(W)]$;*

- *if $t \geq d$, $Z_{t-d+1}$ (and thus $c_{t-d+1}$) is revealed to the learner.*

The main difference between our version of the generalization game and the standard one of Lugosi and Neu (2023) is the introduction of a *delay* on the online learning algorithm's decisions. Specifically, we will force the online learner to only take information into account up to time $t - d$ when picking their action $P_t$. Clearly, setting $d = 1$ recovers the original version of the generalization game with no delay.

It is easy to see that the regret decomposition of Theorem 1 still remains valid in the current setting. The purpose of introducing the delay is to be able to make sure that the term $M_n = -\frac{1}{n} \sum_{t=1}^n \langle P_t, c_t \rangle$ is small. The lemma below states that the increments of $M_n$ behave similarly to a martingale-difference sequence, thanks to the introduction of the delay.

**Lemma 1.** *Fix $d \in [\![1, n]\!]$. Under assumption 1, it holds for all $t \in [\![1, n]\!]$:*

$$\mathbb{E}[\langle -P_t, c_t \rangle | \mathcal{F}_{t-d}] \leq \phi_d .$$

*where $P_t$ and $c_t$ are defined as in 1.*

*Proof.* Since $P_t$ is $\mathcal{F}_{t-d}$-measurable we have $\mathbb{E}[\langle -P_t, c_t \rangle | \mathcal{F}_{t-d}] = \langle P_t, \mathbb{E}[-c_t | \mathcal{F}_{t-d}] \rangle \leq \phi_d$, where the last step uses Assumption 1. $\qquad \square$

Thus, by following the decomposition of Theorem 1, we are left with the problem of bounding the regret of the delayed online learning algorithm against $P_{W_n|S_n}$, denoted as $\text{Regret}_{d,n}(P_{W_n|S_n}) = \sum_{t=1}^n \langle P_t - P_{W_n|S_n}, c_t \rangle$. The following proposition states a simple and clean bound that one can immediately derive from these insights.

**Proposition 1** (Bound in expectation). *Consider $(Z_t)_{t \in \mathbb{N}^*}$ satisfying Assumption 1 and suppose there exists a $d$-delayed online learning algorithm with regret bounded by $\text{Regret}_{d,n}(P^*)$ against any comparator $P^*$. Then, the expected generalization of $\mathcal{A}$ is bounded as*

$$\mathbb{E}\left[\text{Gen}(\mathcal{A}, S_n)\right] \leq \frac{\mathbb{E}\left[\text{Regret}_{d,n}(P_{W_n|S_n})\right]}{n} + \phi_d .$$

*Proof.* By Theorem 1, it holds that $\mathbb{E}[\text{Gen}(\mathcal{A}, S_n)] = \frac{\mathbb{E}[\text{Regret}_{d,n}(P_{W_n|S_n})]}{n} + \mathbb{E}[M_n]$, where the regret is for a strategy $P_t$ in the delayed generalization game. Hence, by Lemma 1

$$\mathbb{E}[M_n] = \mathbb{E}\left[-\frac{1}{n}\sum_{t=1}^n \langle P_t, c_t \rangle\right] = \frac{1}{n}\sum_{t=1}^n \mathbb{E}[\langle -P_t, c_t \rangle] = \frac{1}{n}\sum_{t=1}^n \mathbb{E}\left[\mathbb{E}[\langle -P_t, c_t \rangle | \mathcal{F}_{t-d}]\right] \leq \phi_d ,$$

which proves the claim. $\qquad \square$

The above result holds in expectation over the training sample. We now provide a high-probability guarantee on the generalization error.

**Theorem 3** (Bound in probability). *Assume that $(Z_t)_{t \in \mathbb{N}^*}$ satisfies Assumption 1 and consider a $d$-delayed online learning algorithm with regret bounded by $R_{d,n}(P^*)$ against any comparator $P^*$. Then, for any $\delta > 0$, it holds with probability $1 - \delta$ on the draw of $S_n$, uniformly for all $\mathcal{A}$,*

$$\text{Gen}(\mathcal{A}, S_n) \leq \frac{R_{d,n}(P_{W_n|S_n})}{n} + \phi_d + \sqrt{\frac{2d \log\left(\frac{d}{\delta}\right)}{n}} .$$

The proof of this claim follows directly from combining the decomposition of Theorem 1 with a standard concentration result for mixing processes that we state below.

**Lemma 2.** *Fix $d \in [\![1, n]\!]$ and consider $(Z_t)_{t \in \mathbb{N}^*}$ satisfying Assumption 1. Consider the generalization game of Definition 1. Then, for any $\delta > 0$, the following bound is satisfied with probability at least $1 - \delta$:*

$$M_n \leq \phi_d + \sqrt{\frac{2d \log\left(\frac{d}{\delta}\right)}{n}}.$$

The proof is based on a classic "blocking" technique due to Yu (1994). For the sake of completeness, we provide a proof in Appendix A.2.

## 4 New generalization bounds for non-*i.i.d.* data

The dependence on the delay $d$ for the bounds that we presented in the previous section is non-trivial. Indeed, if on the one hand increasing the delay will reduce the magnitude of $\phi_d$, on the other hand the regret of the online learner will grow with $d$. There is hence a trade-off between these two terms appearing in our bounds. In what follows, we derive some concrete generalization bounds from Theorem 3, under a number of different choices of the online learning algorithm. For concreteness, we will consider two types of mixing assumptions, but stress that the approach can be applied to any process that satisfies Assumption 1.

### 4.1 Regret bounds for delayed online learning

From Theorem 3, we can obtain a generalization bound using our framework if we have a regret bound for a delayed online algorithm. This is a well-known problem in the area of online learning (see, *e.g.*, Weinberger and Ordentlich, 2002; Joulani et al., 2013). In the following, we will leverage the following simple trick that allows us to extend the regret bounds of any online learning algorithm to its delayed counterpart, provided that the regret bound respects some specific assumptions.

**Lemma 3** (Weinberger and Ordentlich, 2002). *Consider any online algorithm whose regret satisfies* $\text{Regret}_n(P^*) \leq R(n)$ *for any comparator $P^*$, where $R$ is a non-decreasing real-valued function such that $y \mapsto yR(x/y)$ is a concave function of $y$ for any fixed $x$. Then, for any $d \geq 1$ there exists an online learning algorithm with delay $d$ such that, for any comparator $P^*$,*

$$\text{Regret}_{d,n}(P^*) \leq dR\left(n/d\right).$$

The proof idea is closely related to the blocking trick of Yu (1994), with an algorithmic construction that runs one instance of the base method for each index $i = 1, 2, \ldots, d$, with the $i$-th instance being responsible for the regret in rounds $i, i + d, i + 2d, \ldots$ (more details are provided in Appendix B.3). For most of the regret bounds that we consider, the function $R$ takes the form $R(n) = O(\sqrt{n})$, so that the first term in the generalization bound is typically of order $\sqrt{d/n}$. Since this term matches the bound on $M_n$ in Lemma 2, in this case the final generalization bound behaves effectively as if the sample size was $n/d$ instead of $n$.

### 4.2 Geometric and algebraic mixing

The following definition gives two concrete examples of mixing processes that satisfy Assumption 1 with different choices of $\phi_d$, and are commonly considered in the related literature (see, e.g., Mohri and Rostamizadeh, 2010, Levin and Peres, 2017).

**Definition 2.** *We say that a stationary process $(Z_t)_{t \in \mathbb{N}^*}$ satisfying Assumption 1 is:*

- *geometrically mixing, if $\phi_d = Ce^{-\frac{d}{\tau}}$, for some positive $\tau$ and $C$;*

- *algebraically mixing, if $\phi_d = Cd^{-r}$, for some positive $r$ and $C$.*

Instantiating the bound of Theorem 3 to these two cases yields the following two corollaries.

**Corollary 1.** *Assume $(Z_t)_{t \in \mathbb{N}^*}$ is a geometrically mixing process with constants $\tau, C > 0$. Consider a $d$-delayed online learning algorithm with regret bounded by $R_{d,n}(P^*)$ for all comparators $P^*$.*

*Then, setting $d = \lceil \tau \log n \rceil$, for any $\delta > 0$, with probability at least $1 - \delta$ we have that, uniformly for any algorithm $\mathcal{A}$,*

$$\mathrm{Gen}(\mathcal{A}, S_n) \leq \frac{R_{d,n}(P_{W_n|S_n})}{n} + \frac{C}{n} + \sqrt{\frac{2\left(\tau \log n + 1\right) \log\left(\frac{\tau \log n + 1}{\delta}\right)}{n}} \,.$$

Up to a term linear in $\tau$ and some logarithmic factors, the above states that under the geometric mixing the same rates are achievable as in the *i.i.d.* setting. Roughly speaking, this amounts to saying that the effective sample size is a factor $\tau$ smaller than the original number of samples $n$, as long as generalization is concerned.

**Corollary 2.** *Assume $(Z_t)_{t \in \mathbb{N}^*}$ is an algebraic mixing process with constants $r, C > 0$. Consider a $d$-delayed online learning algorithm with regret bounded by $R_{d,n}(P^*)$ against any comparator $P^*$. Then, setting $d = \left(C^2 n\right)^{1/(1+2r)}$, for any $\delta > 0$, with probability at least $1 - \delta$ we have that, uniformly for any algorithm $\mathcal{A}$,*

$$\mathrm{Gen}(\mathcal{A}, S_n) \leq \frac{R_{d,n}(P_{W_n|S_n})}{n} + C\left(1 + \sqrt{\log(d/\delta)}\right) n^{-\frac{2r}{2(1+2r)}} \,.$$

This result suggests that the rates achievable for algebraically mixing processes are qualitatively much slower than what one can get for *i.i.d.* or geometrically mixing data sequences (although the rates do eventually approach $1/\sqrt{n}$ as $r$ goes to infinity).

### 4.3 Multiplicative weights with delay

We start our discussion on possible online strategies by focusing on the classic exponential weighted average (EWA) algorithm (Vovk, 1990; Littlestone and Warmuth, 1994; Freund and Schapire, 1997). We fix a data-free prior $P_1 \in \Delta_{\mathcal{W}}$ and a learning rate parameter $\eta > 0$. We consider the updates

$$P_{t+1} = \arg\min_{P \in \Delta_{\mathcal{W}}} \left\{ \langle P, c_t \rangle + \frac{1}{\eta} \mathcal{D}_{KL}(P || P_t) \right\},$$

Combining the standard regret bound of EWA (see Appendix B.1) with Lemma 3 and Corollary 1 yields the result that follows.

**Corollary 3.** *Suppose that $(Z_t)_{t \in \mathbb{N}^*}$ is a geometric mixing process with constants $\tau, C > 0$. Suppose that $\ell(w, z) \in [0, 1]$ for all $w, z$. Then, for any $P_1 \in \Delta_{\mathcal{W}}$ and any $\delta > 0$, with probability at least $1 - \delta$, uniformly on any learning algorithm $\mathcal{A}$ we have*

$$\mathrm{Gen}(\mathcal{A}, S_n) \leq \frac{\mathcal{D}_{KL}(P^* || P_1)(\tau \log n + 1)}{\eta n} + \frac{\eta}{2} + \frac{C}{n} + \sqrt{\frac{2\left(\tau \log n + 1\right) \log\left(\frac{\tau \log n + 1}{\delta}\right)}{n}} \,.$$

This results suggests that when considering geometric mixing processes, by applying a union bound over a well-chosen range of $\eta$ we recover the PAC-Bayes bound of McAllester (1998) up to a $O(\sqrt{\tau \log n})$ factor. A similar result can be derived from Corollary 2 for algebraically mixing processes, leading to a bound typically scaling as $n^{-2r/(2(1+2r))}$.

### 4.4 Follow the regularized leader with delay

In this subsection we extend the common class of online learning algorithms known as follow the regularized leader (FTRL, see *e.g.*, Abernethy and Rakhlin, 2009; Orabona, 2019) to the problem of learning with delay. FTRL algorithms are defined using a convex regularization function $h : \Delta_{\mathcal{W}} \to \mathbb{R}$. We restrict ourselves to the set of proper, lower semi-continuous and $\alpha$-strongly convex functions with respect to a norm $||.||$ (and its respective dual norm $||.||_*$) defined on the set of signed finite measures on $\mathcal{W}$ (see Appendix B.2 for more details). The online procedure (without delay) of the FTRL algorithm is as follows:

$$P_{t+1} = \arg\min_{P \in \Delta_{\mathcal{W}}} \left\{ \sum_{s=1}^{t} \langle P, c_s \rangle + \frac{1}{\eta} h(P) \right\} \,.$$

The existence of the minimum is guaranteed by the compactness of $\Delta_{\mathcal{W}}$ under $\|\cdot\|$, and its uniqueness is ensured by the strong convexity of $h$. Combining the analysis of FTRL (see Appendix B.2) with Lemma 3 and Corollary 1 yields the following result.

**Corollary 4.** *Suppose that $(Z_t)_{t \in \mathbb{N}^*}$ is a geometric mixing process with constants $\tau, C > 0$. Suppose that $\ell(w, z) \in [0, 1]$ for all $w, z$. Assume there exists $B > 0$ such that for all $t$, $\|c_t\|_* \leq B$. Then, for any $P_1 \in \Delta_{\mathcal{W}}$, for any $\delta > 0$ with probability at least $1 - \delta$ on the draw of $S_n$, uniformly for all $\mathcal{A}$,*

$$\mathrm{Gen}(\mathcal{A}, S_n) \leq \frac{(h(P^*) - h(P_1))\,(\tau \log n + 1)}{\eta n} + \frac{\eta B^2}{2\alpha} + \frac{C}{n} + \sqrt{\frac{2\,(\tau \log n + 1)\log\left(\frac{\tau \log n + 1}{\delta}\right)}{n}}.$$

This generalization bound is similar to the bound of Theorem 9 of Lugosi and Neu (2023) up to a $O(\sqrt{\tau \log n})$ factor, when applying a union-bound argument over an appropriate grid of learning-rates $\eta$. In particular, this result recovers PAC-Bayesian bounds like those of Corollary 3 when choosing $h = \mathcal{D}_{\mathrm{KL}}\left(\cdot \| P_1\right)$. We refer to Section 3.2 in Lugosi and Neu (2023) for more discussion on such bounds. As before, a similar result can be stated for algebraically mixing processes, with the leading terms approaching zero at rate of $n^{-2r/2(1+2r)}$ instead of $n^{-1/2}$.

# 5 Generalization bounds for dynamic hypotheses

Finally, inspired by the works of Eringis et al. (2022, 2024), we extend our framework to accommodate loss functions $\ell$ that rely not only on the last data point $Z_t$, but on the entire data sequence $\overline{Z}_t = (Z_t, Z_{t-1}, \ldots, Z_1)$. Formally, we will consider loss functions of the form $\ell : \mathcal{W} \times \mathcal{Z}^* \to \mathbb{R}_+$[1] and write $\ell(w, \overline{z}_t)$ to denote the loss associated with hypothesis $w \in \mathcal{W}$ on sequence $\overline{z}_t \in \mathcal{Z}^t$. This consideration extends the learning problem to class of dynamical predictors such as Kalman filters, autoregressive models, or recurrent neural networks (RNNs), broadly used in time-series forecasting (Ariyo et al., 2014; Takeda et al., 2016). Specifically, if we think of $z_t = (x_t, y_t)$ as a data-pair of context and observation, in time-series prediction we usually not only rely on the context $x_t$ but also on the past sequence of contexts and observations $(x_{t-1}, y_{t-1}, \ldots, x_1, y_1)$. As an example, consider $\ell(w, z_t, \ldots, z_1) = \frac{1}{2}(y_t - h_w(x_t, z_{t-1}, \ldots, z_1))^2$ where $h \in \mathcal{H}$ is a function class parameterized by $\mathcal{W}$. For this type of loss function a natural definition of the test error is:

$$\widetilde{\mathcal{L}}(w) = \lim_{n \to \infty} \mathbb{E}[\ell(w, Z'_t, Z'_{t-1}, \ldots, Z'_{t-n})],$$

where $\overline{Z}'_t = (Z'_t, Z'_{t-1}, \ldots)$ is a semi-infinite random sequence drawn from the same stationary process that has generated the data $\overline{Z}_t$. We consider the following assumption.

**Assumption 2.** *For a given process $(Z_t)_{t \in \mathbb{Z}}$ with joint-distribution $\nu$ over $\mathcal{Z}^{\mathbb{Z}}$ and same marginals $\mu$ over $\mathcal{Z}$, there exists a non-increasing sequence $(\phi_d)_{d \in \mathbb{N}^*}$ of non-negative real numbers such that the following holds for all $w \in \mathcal{W}$, for all $t \in \mathbb{N}^*$:*

$$\mathbb{E}\left[\ell(w, Z_t, \ldots, Z_1) - \widetilde{\mathcal{L}}(w)\Big|\mathcal{F}_{t-d}\right] \leq \phi_d.$$

This is a generalization of Assumption 1 in the sense that taking $\ell(w, Z_t, \ldots, Z_1) = \ell(w, Z_t)$ simply amounts to requiring the same mixing condition as before. For our online-to-PAC conversion we consider the same framework as in Definition 1, except that now the cost function is defined as

$$c_t : w \mapsto \ell(w, Z_t, \ldots, Z_1) - \widetilde{\mathcal{L}}(w)\,.$$

Then it easy to check that result of Lemma 2 still holds for this specific cost, and we can thus extend all the results of Section 4. For concreteness, we state the following adaptation of Theorem 3 below.

**Theorem 4.** *Assume $(Z_t)_{t \in \mathbb{Z}}$ which satisfies Assumption 2 and consider a $d$-delayed online learning algorithm with regret bounded by $R_{d,n}(P^*)$ against any comparator $P^*$. Then, for any $\delta > 0$, it holds with probability $1 - \delta$:*

$$\mathrm{Gen}(\mathcal{A}, S_n) \leq \frac{R_{d,n}(P_{W_n|S_n})}{n} + \phi_d + \sqrt{\frac{2d \log\left(\frac{d}{\delta}\right)}{n}}.$$

---

[1]Here, $\mathcal{Z}^*$ denotes the disjoint union $\mathcal{Z}^* = \sqcup_{t \in \mathbb{N}} \mathcal{Z}^t$.

To see that Assumption 2 can be verified and the resulting bounds can be meaningfully applied, consider the following concrete assumptions about the hypothesis class, the loss function, and the data generating process. The first assumption says that for any given hypothesis, the influence of past data points on the associated loss vanishes with time (*i.e.*, the hypothesis forgets the old data points at a controlled rate).

**Assumption 3.** *There exists a decreasing sequence $(B_d)_{d \in \mathbb{N}^*}$ of non-negative real numbers such that for any two sequences $\overline{z}_t = (z_t, \ldots, z_i)$ and $\overline{z}'_t = (z'_t, \ldots, z'_j)$ of possibly different lengths that satisfy $z_k = z'_k$ for all $k \in t, \ldots, t - d + 1$, we have $|\ell(w, \overline{z}_t) - \ell(w, \overline{z}'_t)| \leq B_d$, for all $w \in \mathcal{W}$.*

This condition can be verified for stable dynamical systems like autoregressive models, certain classes of RNNs, or sequential predictors that have bounded memory by design (see Eringis et al., 2022, 2024). The next assumption is a refinement of Assumption 1, adapted to the case where the loss function acts on blocks of $d$ data points $\overline{z}_{t-d+1:t} = (z_t, z_{t-1}, \ldots, z_{t-d+1})$.

**Assumption 4.** *Let $\overline{Z}_t = (Z_t, \ldots, Z_1)$ be a sequence of data points and let $\overline{Z}'_t = (Z'_t, \ldots, Z'_0, \ldots)$ be an independent copy of the same process. Then, there exists a decreasing sequence $(\beta_d)_{d \in \mathbb{N}^*}$ non-negative real numbers such that the following is satisfied for all hypotheses $w \in \mathcal{W}$ and all $d \in \mathbb{N}^*$:*

$$\mathbb{E}\left[\ell(w, \overline{Z}'_{t-d+1:t}) - \ell(w, \overline{Z}_{t-d+1:t}) \middle| \mathcal{F}_{t-2d}\right] \leq \beta_d.$$

This assumption can be verified whenever the loss function is bounded and the joint distribution of the data block $\overline{Z}_{t-d+1:t}$ satisfies a $\beta$-mixing assumption. In more detail, this latter condition amounts to requiring that the conditional distribution of each data block given a block that trails $d$ steps behind is close to the marginal distribution in total variation distance, up to an additive term of $\beta_d$. The following proposition shows that these two simple conditions together imply that Assumption 2 holds, and that thus the bound of Theorem 4 can be meaningfully instantiated for bounded-memory hypothesis classes deployed on mixing processes.

**Proposition 2.** *Suppose that the loss function satisfies Assumption 3 and the data distribution satisfies Assumption 4. Then Assumption 2 is satisfied with $\phi_d = 2B_{d/2} + \beta_{d/2}$.*

# 6 Conclusion

We have developed a general framework for deriving generalization bounds for non-i.i.d. processes under a general mixing assumption, via an extension of the online-to-PAC-conversion framework of Lugosi and Neu (2023). Among other results, this approach has allowed us to prove PAC-Bayesian generalization bounds for such data in a clean and transparent way, and even study classes of dynamic hypotheses under a simple bounded-memory condition. These results provide a clean and tight alternative to the results of (Alquier and Wintenberger, 2012; Eringis et al., 2022). The generality of our approach further demonstrates the power of the Online-to-PAC scheme of Lugosi and Neu (2023), and in particular our results provide further evidence that this framework is particularly promising for developing techniques for generalization in non-i.i.d. settings. We hope that flexibility of our framework will find further uses and enables more rapid progress in the area.

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

 # A  Omitted proofs

 ## A.1  The proof of Theorem 1

 Let $(P_t)_{t=1}^n \in \Delta_{\mathcal{W}}^n$ be the predictions of an online learner playing the *generalization game*. Then

$$
\begin{aligned}
\mathrm{Gen}(\mathcal{A}, S_n) &= \frac{1}{n} \sum_{t=1}^n \mathbb{E}[\ell_t(W_n) - \mathcal{L}(W_n)|S_n] \\
&= -\frac{1}{n} \sum_{t=1}^n \mathbb{E}[c_t(W_n)|S_n] \\
&= -\frac{1}{n} \sum_{t=1}^n \langle P_{W_n|S_n}, c_t \rangle \\
&= \frac{1}{n} \sum_{t=1}^n \langle P_t - P_{W_n|S_n}, c_t \rangle - \frac{1}{n} \sum_{t=1}^n \langle P_t, c_t \rangle \\
&= \frac{\mathrm{Regret}_n(P_{W_n|S_n})}{n} + M_n.
\end{aligned}
$$

 ## A.2  The proof of Lemma 2

 Assume $n = Kd$ for simplicity:

$$
\begin{aligned}
M_n &= -\frac{1}{n} \sum_{t=1}^n \langle P_t, c_t \rangle \\
&= \frac{1}{dK} \sum_{i=1}^d \sum_{t=1}^K \langle -P_{i+d(t-1)}, c_{i+d(t-1)} \rangle
\end{aligned}
$$

 We denote $X_t^{(i)} = \langle -P_{i+d(t-1)}, c_{i+d(t-1)} \rangle$ and we want to bound in high-probability the term
 $\frac{1}{K} \sum_{t=1}^K X_t^{(i)}$. Let also denote $\mathcal{F}_t^{(i)} = \mathcal{F}_{i+d(t-1)}$. Then for $i \in [\![1, d]\!]$, we can write using Chernoff's
 technique that for all $\lambda > 0$ it holds:

$$
\begin{aligned}
\mathbb{P}\left( \frac{1}{K} \sum_{t=1}^K X_t^{(i)} \geq u \right) &\leq \frac{\mathbb{E}\left[ e^{\frac{\lambda}{K} \sum_{t=1}^K X_t^{(i)}} \right]}{e^{\lambda u}} \\
&\leq \mathbb{E}\left[ e^{\frac{\lambda}{K} \sum_{t=1}^{K-1} X_t^{(i)}} \mathbb{E}\left[ e^{\frac{\lambda}{K} X_K^{(i)}} \middle| \mathcal{F}_{K-1}^{(i)} \right] \right] e^{-\lambda u}.
\end{aligned}
$$

 Now remark that:

$$
\mathbb{E}\left[ e^{\frac{\lambda}{K} X_K^{(i)}} \middle| \mathcal{F}_{K-1}^{(i)} \right] = \mathbb{E}\left[ e^{\frac{\lambda}{K} (X_K^{(i)} - \mathbb{E}[X_K^{(i)}|F_{K-1}^{(i)}])} \middle| F_{K-1}^{(i)} \right] e^{\frac{\lambda}{K} \mathbb{E}[X_K^{(i)}|F_{K-1}^{(i)}]}.
$$

 If we denote $Z = X_K^{(i)} - \mathbb{E}[X_K^{(i)}|F_{K-1}^{(i)}]$ then $|Z| \leq 2$ and $\mathbb{E}[Z|F_{K-1}^{(i)}] = 0$ so via Hoeffding's
 lemma:

$$
\mathbb{E}[e^{\frac{\lambda}{K} Z}] \leq e^{\frac{\lambda^2}{2K^2}}.
$$

 Now by construction of the $P_t$ and because of Lemma 1 it follows that for all $i$, $\mathbb{E}[X_K^{(i)}|F_{K-1}^{(i)}] \leq \phi_d$.
 Repeating the same reasoning for each term of the sum yields:

$$
\mathbb{P}\left( \frac{1}{K} \sum_{t=1}^K X_t^{(i)} \geq u \right) \leq e^{\frac{\lambda^2}{2K}} e^{\lambda \phi_d} e^{-\lambda u}.
$$

Optimzing with $\lambda = K(u - \phi_d)$ and taking $\delta = e^{-\frac{K(u-\phi)^2}{2}}$ it finally holds for any $\delta > 0$, with probability $1 - \frac{\delta}{d}$:

$$
\frac{1}{K} \sum_{t=1}^K X_t^{(i)} \leq \phi_d + \sqrt{\frac{2 \log\left( \frac{d}{\delta} \right)}{K}}.
$$

Thus applying a union bound we have with probability $1 - \delta$:

$$M_n \leq \phi_d + \sqrt{\frac{2\log\left(\frac{d}{\delta}\right)}{K}},$$

which concludes the proof. $\qquad\square$

### A.3 Proof of Proposition 2

Suppose without loss of generality that $d$ is even and define $d' = d/2$. For the proof, let $\overline{Z}'_n$ be a semi-infinite sequence drawn independently from the same process as $\overline{Z}_n$. Then, we have

$$
\begin{aligned}
\tilde{\mathcal{L}}(w) &= \lim_{n\to\infty} \mathbb{E}[\ell(w, Z'_t, Z'_{t-1}, ..., Z'_{t-n})] \\
&\leq \mathbb{E}[\ell(w, Z'_t, Z'_{t-1}, \ldots, Z'_{t-d'})] + B_{d'} \\
&\leq \mathbb{E}\left[\ell(w, Z_t, Z_{t-1}, \ldots, Z_{t-d'})\,|\,\mathcal{F}_{t-2d'}\right] + B_{d'} + \beta_{d'} \\
&\leq \mathbb{E}\left[\ell(w, Z_t, Z_{t-1}, \ldots, Z_{t-d'}, \ldots, Z_1)\,|\,\mathcal{F}_{t-2d'}\right] + 2B_{d'} + \beta_{d'} \\
&\leq \mathbb{E}\left[\ell(w, Z_t, Z_{t-1}, \ldots, Z_1)\,|\,\mathcal{F}_{t-2d'}\right] + 2B_{d'} + \beta_{d'},
\end{aligned}
$$

where we used Assumption 3 in the first inequality, Assumption 4 in the second one, and Assumption 3 again in the last step. This proves the statement. $\qquad\square$

## B  Online Learning Tools and Results

### B.1  Regret Bound for EWA

Recalling EWA updates we have:

$$P_{t+1} = \arg\min_{P\in\Delta_{\mathcal{W}}} \left\{ \langle P, c_t\rangle + \frac{1}{\eta}\mathcal{D}_{KL}(P||P_t)\right\},$$

where $\eta > 0$ is a learning-rate parameter. The minimizer can be shown to exist and satisfies:

$$\frac{\mathrm{d}P_{t+1}}{\mathrm{d}P_t}(w) = \frac{e^{-\eta c_t(w)}}{\int_{\mathcal{W}} e^{-\eta c_t(w')}\mathrm{d}P_t(w')},$$

and the following result holds.

**Proposition 3.** *For any prior $P_1 \in \Delta_{\mathcal{W}}$ and any comparator $P^* \in \Delta_{\mathcal{W}}$ the regret of EWA simultaneously satisfies for $\eta > 0$:*

$$\mathrm{Regret}(P^*) \leq \frac{\mathcal{D}_{KL}(P^*||P_1)}{\eta} + \frac{\eta}{2}\sum_{t=1}^{n}||c_t||_\infty^2.$$

We refer the reader to Appendix A.1 of Lugosi and Neu (2023) for a complete proof of the result above.

### B.2  Regret Bound for FTRL

We say that $h$ is $\alpha-$strongly convex if the following inequality is satisfied for all $P, P' \in \Delta_{\mathcal{W}}$ and all $\lambda \in [0, 1]$:

$$h(\lambda P + (1-\lambda)P') \leq \lambda h(P) + (1-\lambda)h(P') - \frac{\alpha\lambda(1-\lambda)}{2}||P - P'||^2.$$

Recalling the FTRL updates:

$$P_{t+1} = \arg\min_{P\in\Delta_{\mathcal{W}}} \left\{\sum_{s=1}^{t}\langle P, c_s\rangle + \frac{1}{\eta}h(P)\right\},$$

the following results holds.

**Proposition 4.** *For any prior $P_1 \in \Delta_{\mathcal{W}}$ and any comparator $P^* \in \Delta_{\mathcal{W}}$ the regret of FTRL simultaneously satisfies for $\eta > 0$:*

$$\text{Regret}_n(P^*) \leq \frac{h(P^*) - h(P_1)}{\eta} + \frac{\eta}{2\alpha} \sum_{t=1}^{n} ||c_t||_*^2.$$

We refer the reader to Appendix A.3 of Lugosi and Neu (2023) for a complete proof of the results above.

### B.3 Details about the reduction of Weinberger and Ordentlich (2002)

For concretenes we formally present how to turn any online learning algorithm into its delayed version. For sake of convenience, assume $n = Kd$. We denote $\tilde{c}_t^{(i)} = c_{i+d(t-1)}$ (for instance $\tilde{c}_1^{(1)} = c_1$ is the cost revealed at time $d + 1$). Then we create $d$ instances of horizon time $K$ of the online learning as follows, for $i = 1, \ldots, d$:

- We initialize $\tilde{P}_1^{(i)} = P_0$,
- for each block $i$ of length $K$ we update for $t = 1, \ldots, K$:

$$\tilde{P}_{t+1}^{(i)} = \text{OL}_{\text{update}}\left( (\tilde{c}_s^{(i)})_{s=1}^t \right).$$

Here $\text{OL}_{\text{update}}$ refers to the update function of the online learning algorithm we consider which can possibly depend of the whole history of cost functions (e.g., in the case of the FTRL update).

