# OpenReview forum: "Generalization bounds for mixing processes via delayed online-to-PAC conversions"
_NeurIPS.cc/2024/Conference — Submitted to NeurIPS 2024_

### Official Review · Reviewer_BFxn · 2024-06-21

**Soundness:** 3
**Presentation:** 4
**Contribution:** 2
**Rating:** 6
**Confidence:** 3

**Summary:**

The authors establish generalization error bounds for non-iid data based on the online-to-PAC conversion.
In particular, the authors extend the online-to-PAC conversion techinique to non-iid settings utilizing the online learning with delayed feedback. In the paper, the authors illustrate

1. a method of non-iid online-to-PAC conversion,
2. a method of converting online learning algorithms to their delayed counterparts,
3. the resulting non-iid generalization bounds combining 1. and 2., and
4. an extension of 1. to dynamic hypothesis learning.

**Strengths:**

- The well-organized, well-motivated paper on generalization error analysis and online learning.
- The claimed results are novel and help us better understand the generalization in dynamic environments.

**Weaknesses:**

- Notable logical gap: Lemma 3 only gives a regret bound **independent** of $P^*$, but it seems Corollaries 3 and 4 need $P^*$-dependent regret bound. I believe this is fixable, but still, need some fix.
- More discussion on related work: Are there any results previously not known, but can be proved with the proposed method?

**Questions:**

- Can you show instantiations of Theorem 4 using EWA and FTRL (as in Corollaries 3 and 4)?

**Limitations:**

Limitations are not explicitly discussed.

---

> ### Author Rebuttal · Authors · 2024-08-05
>
> Q1) Notable logical gap: Lemma 3 only gives a regret bound independent of $P^*$, but it seems Corollaries 3 and 4 need $P^*$-dependent regret bound. I believe this is fixable, but still, need some fix.
>
> Thanks for pointing this out! This typo is indeed a bit confusing: the right hand side is of course also allowed to depend on $P^*$, and it indeed should depend on the comparator for the result to be applicable. We will fix this in the final version.
>
> Q2) More discussion on related work: Are there any results previously not known, but can be proved with the proposed method?
>
> The explicit bounds that we propose in corollary 3 and 4 are all novel, and our mixing assumptions are generally weaker than what is usually considered in the literature. We will emphasize this more effectively in the final version.
>
> Q3) Can you show instantiations of Theorem 4 using EWA and FTRL (as in Corollaries 3 and 4)?
>
> We omitted these instantiations due to space limitations, otherwise they can easily be derived by analogy with the two corollaries you mention. We will expand on this, either by adding explicit examples or by mentioning them more explicitly.

---

> > ### Comment · Reviewer_BFxn · 2024-08-08
> >
> > Thanks for the rebuttal.
> > I confirm my concerns/questions are all addressed adequately.

---

### Official Review · Reviewer_yvPp · 2024-07-09

**Soundness:** 3
**Presentation:** 3
**Contribution:** 3
**Rating:** 5
**Confidence:** 3

**Summary:**

This paper studies the generalization error of statistical learning in a non-i.i.d setting, where the training data distribution could have temporal dependency. They develop a framework that reduces the generalization error in this case into the regret of an online learning problem with delayed feedback. Then, they present a series of instantiations of their results with different online learning algorithms and assumptions on the data generation process.

**Strengths:**

1. The propose framework is elegant and wildly applicable to many real-world data generating process.
2. The paper is easy to follow, and the setting is well presented. The introductory section for the reduction in the i.i.d case is very helpful in understanding the context.

**Weaknesses:**

1. The proposed framework seems to be a straightforward extension of that in the i.i.d setting. Technical novelty of this work seems limited.
2. The instantiation of the framework given in this paper is still very high level and abstract (for example, the algorithm considered is the general follow the leader algorithm). It would be beneficial to have some specific instantiations and show that if the obtained results are comparable to the existing ones, similar to what has been done in Lugosi and Neu (2023).

**Questions:**

1. typo in Line 43: double "propose"
2. Assuming that the data is $\beta$-mixing, how is your results compare to previous work? Does your result improve over existing ones?
3. Is the framework applicable when the online learning algorithm is under the online mirror descent framework?

**Limitations:**

Nothing necessary stands out.

---

> ### Author Rebuttal · Authors · 2024-08-05
>
> Regarding the technical novelty of our method, please see our general response.
>
> Regarding instantiating our results for some specific settings: We omitted these instantiations due to space limitations, otherwise they can easily be derived by analogy with results of Lugosi and Neu (2023). We will expand on this, either by adding explicit examples or at least by mentioning them more explicitly.
>
> Q1) Assuming that the data is 𝛽-mixing, how is your results compare to previous work? Does your result improve over existing ones?
>
> Our assumption is weaker when the loss is bounded (hence implied by the beta-mixing condition in the papers you mention). For unbounded losses, neither of the assumptions is stronger than the other (i.e., one may have beta mixing without our assumption being satisfied in some cases, but in other cases our assumption may be satisfied without beta mixing). The results provided in our work improve over those of Mohri (2008): those involve a Rademacher complexity term that is often looser in practice than what one can derive from PAC-Bayesian bounds.
>
> Q2) Is the framework applicable when the online learning algorithm is under the online mirror descent framework?
>
> Our framework is fully general and can make use of any online learning algorithm with bounded regret. In particular, one can use OMD-style algorithms through the reduction stated in Section 4.1, and obtain results that are essentially identical to the results stated in Section 4.4 for FTRL-style methods. (The only difference would be replacing $h(P^*) - h(P_1)$ by the Bregman divergence $B_h(P^*\| P_1)$, which are equal when $P_1$ is chosen as the minimizer of $h$.) We refer to Sections 6 and 7 in “A Modern Introduction to Online Learning” by Orabona (2019) for a detailed discussion of both families of methods.

---

### Official Review · Reviewer_DR73 · 2024-07-10

**Soundness:** 3
**Presentation:** 4
**Contribution:** 2
**Rating:** 4
**Confidence:** 5

**Summary:**

This paper extends the Lugosi-Neu(2023) framework for upper bounding the generalization error of statistical learning algorithms to the non-i.i.d. setting, by considering that the training samples are drawn from a suitably mixing stochastic process. They show that the existence of a delayed online learner with bounded regret in the Online-to-Batch game of Lugosi-Neu(2023) against an offline learner implies that the offline learner has low generalization error even when trained on data drawn from a mixing stochastic process. The authors also investigate settings such as FTRL and MWU under this model.

**Strengths:**

The paper addresses an important question - How to bound generalization error of statistical learning algorithms trained on non-i.i.d. data in a manner which is independent of the complexity of the statistical learner? The paper does a fine job at establishing the notion of such bounds and some conditions under which such bounds are recoverable.

**Weaknesses:**

The techniques seem to be largely an amalgamation of several papers which have refined the "blocking technique" in various settings, and the key observation that the introduction of delay in online games lead to the online cost being a sum of martingale difference sequences, which essentially allows them to use proof techniques of Lugosi-Neu(2003).

The delayed online learning setting is new to me, and I am not sure how to evaluate its significance versus the standard online learning setting. In fact, it seems like getting similar bounds w.r.t. the standard online setting would involve significant more technical novelty, compared to the current setting.

The stochastic process also seems to be quite well-behaved in comparison to previous works in the literature such as Mohri and Rostamizadeh (2011), who give generalization bounds (in the pure offline setting) under stochastic processes with weaker notions of convergence.

The authors mention that the results hold for a specific class of bounded loss functions, but I could not find specific details regarding this point afterwards in the paper.

**Questions:**

1. Can the authors comment on the delayed learning setup vs the normal online setup of Lugosi-Neu (2003)? Given that this is a central construct essential to the proof idea, I strongly believe that the discussion section should some attention to this question.
2. Is it possible to get high probability bounds for the generalization error when the offline learner is trained on $\beta$-mixing processes as defined in Yu (1994), Meir (2000), Mohri and Rostamizadeh (2008), etc.?
3. Is it possible to get similar results for a more general family of loss functions (for example Lipschitz loss functions)?

**Limitations:**

The discussions of limitations in the current work is limited, and lacks discussion as to why certain choices were made (or overlooked).

---

> ### Author Rebuttal · Authors · 2024-08-05
>
> Q1) Can the authors comment on the delayed learning setup vs the normal online setup of Lugosi-Neu (2003)?
>
> See our general response to all reviewers regarding the necessity / usefulness of delays in this setting. The setting of online learning with delays is well-studied, and the results we borrow from Weinberger & Ordentlich (2002) are minimax optimal. We will expand our discussion of this setting in the final version.
>
> Q2) Is it possible to get high probability bounds for the generalization error when the offline learner is trained on 𝛽-mixing processes as defined in Yu (1994), Meir (2000), Mohri and Rostamizadeh (2008), etc.?
>
> Our assumption is weaker when the loss is bounded (hence implied by the beta-mixing condition in the papers you mention). For unbounded losses, neither of the assumptions is stronger than the other (i.e., one may have beta mixing without our assumption being satisfied in some cases, but in other cases our assumption may be satisfied without beta mixing). The results provided in our work improve over those of Mohri and Rostamizadeh (2008): those involve a Rademacher complexity term that is often looser in practice than what one can derive from PAC-Bayesian bounds.
>
> Q3) Is it possible to get similar results for a more general family of loss functions (for example Lipschitz loss functions)?
>
> We can instantiate more examples, essentially all the settings discussed in Lugosi  and Neu (2022, 2023), with different choices of the convex functional on the measure space. For instance, note that their Section 3.2 lists numerous generalization bounds, some of which hold for unbounded loss functions. All of these can be instantiated in our setting as well, thanks to the generality of our framework. We will emphasize this in the final version.

---

> > ### Comment · Reviewer_DR73 · 2024-08-08
> >
> > Thank you for the rebuttal. I am mostly satisfied with the responses. I will be maintaining my current positive rating.
> >
> > As an aside, I would urge the authors to add an exposition on the delayed setting due to importance of the technique in this paper. It might be well-studied (as the authors have claimed) but I do not think it is well-established to the point of being self-explanatory.

---

### Official Review · Reviewer_saej · 2024-07-11

**Soundness:** 4
**Presentation:** 3
**Contribution:** 4
**Rating:** 7
**Confidence:** 3

**Summary:**

The paper focuses on learning from non-i.i.d data. Specifically, the authors develop a framework that derives generalization guarantees through a reduction to an online learning game with delays, where achieving low regret translates to low generalization error. They present specific bounds when using EWA and FTRL as the online learning algorithms. Additionally, the framework is extended to accommodate dynamic hypotheses.

**Strengths:**

The paper is well-written and easy to follow. The proposed framework is general, novel, and elegantly designed, facilitating a clear translation between low regret in online learning algorithms and low generalization error in the context of mixing data. I appreciate the simplicity and flexibility of the framework, and overall, it represents a valuable contribution to the field

**Weaknesses:**

While I did not go over the entire details, I did not find any major weaknesses.
- A small typo is line 43: "..we propose propose.. "

**Questions:**

Could the authors provide their perspective on potential future directions and limitations of their framework?

---

> ### Author Rebuttal · Authors · 2024-08-05
>
> Q1) Could the authors provide their perspective on potential future directions and limitations of their framework?
>
> We believe this framework and its flexibility should motivate the investigation of generalization bounds in more general non-i.i.d. settings. There are still many questions  not covered in this paper such as the ones raised by other reviewers about considering different assumptions on the mixing process. One limitation of our framework is that it is limited to non-i.i.d. processes that are stationary — but this assumption is necessary to make sure that the test error and the generalization error is well-defined in the first place. In our view, defining notions of generalization without stationarity is the most interesting challenge for further research in this area. We hope that our work can provide interesting insights that may contribute towards achieving this goal.

---

### Official Review · Reviewer_dxR2 · 2024-07-12

**Soundness:** 3
**Presentation:** 4
**Contribution:** 3
**Rating:** 7
**Confidence:** 3

**Summary:**

This paper provides a framework for proving generalization bounds for non-i.i.d. data sequences, building upon a recent framework introduced by Lugosi and Neu (2023) that reduces PAC to online learning. This technique recovers some known PAC Bayesian bounds for non-i.i.d. scenarios and various other implications.

The original framework by Lugosi and Neu (2023) introduced an online learning "generalization game", where the regret of the online learning algorithm can be translated into a generalization bound in the offline setting. This framework has been shown to recover some important generalization bounds with a clean analysis.

In this paper, the generalization game is extended to a game where the learner gets to see the observation with delay (where there's no delay we are back to the original framework). The regret of the online learner in this game can again be translated into a generalization bound. When the delay is large, it increases the regret of the online learner (one term in the generalization bound) but decreases the term determined by the property of "how much the sequence is non-i.i.d.".
Online learning with delays has been studied extensively, and so "off-the-shelf" algorithms and regret bounds can be used to derive/recover generalization bounds.

One nice application of the technique is that it allows the analysis of stationary mixing processes that have been studied extensively (the assumption on the non-i.i.d. sequences is weaker than known mixing assumptions). Another interesting application is to popular dynamic predictions such as autoregressive models and RNNs.

**Strengths:**

Deriving generalization bounds for non-i.i.d. settings is a central effort in the machine learning community and is of great interest. The framework suggested in this paper allows us to do so in a very clean way and might be useful for more applications.

Also, the assumption on the non-i.i.d. sequences is quite weak, which is an advantage.

**Weaknesses:**

The paper heavily builds on the framework of Lugosi and Neu (2023). This is not a weakness, but my question is: besides extending the online game to accommodate delays and using ideas from the online learning literature, what are the technical challenges/contributions in this paper?

Another question - do you know if the paper by Lugosi and Neu (2023) was already published? I'm asking since I didn't go over the proofs in this paper.

**Questions:**

See above.

**Limitations:**

Limitations are properly addressed.

---

> ### Author Rebuttal · Authors · 2024-08-05
>
> Q1) besides extending the online game to accommodate delays and using ideas from the online learning literature, what are the technical challenges/contributions in this paper?
>
> See our general response to all reviews above.
>
> Q2) Do you know if the paper by Lugosi and Neu (2023) was already published? I'm asking since I didn't go over the proofs in this paper?
>
> To our knowledge, that paper is under review at a journal. We note that the proofs in our submission are almost entirely self-contained, and the only really important technical result we use from Lugosi & Neu (2023) is Theorem 1, whose simple proof we reproduce in our own Appendix. The rest of the results we cite from that work are mostly standard regret bounds that can be found in many other references on online learning.

---

> > ### Comment · Reviewer_dxR2 · 2024-08-12
> >
> > Thanks for your response.
> > I will keep my positive score.

---

### Author Rebuttal · Authors · 2024-08-05

We thank the reviewers for their time and constructive feedback on our submission, which we will incorporate to improve our paper. We are glad to see that all reviewers have appreciated our contribution, and in particular the simplicity and generality of our framework.

Some reviewers have asked about the technical novelty of our results. In response, let us say that the technical contribution is really as simple as introducing delays into the online learning algorithm to deal with non-stationarity. We can see why this idea may feel natural in hindsight, we wish to take this opportunity to emphasize that it was not obvious a priori that such a simple idea would solve the problem we study in the paper. In fact, after the NeurIPS submission deadline, a concurrent paper appeared on arxiv studying the exact same problem, but without making use of delays:
https://arxiv.org/abs/2405.13666
Their analysis is directly inspired by the work of Agarwal and Duchi (2011) on online-to-batch conversions for convex optimization, and required much stronger assumptions than our analysis based on delays. While we still think that this arxiv paper presents an interesting contribution, we believe that it also nicely illustrates how non-trivial and useful the idea of introducing delays into the online-to-PAC framework is. We hope that the reviewers will find this response to be helpful in assessing the value of our contribution.

---

### Decision · Program_Chairs · 2024-09-25

**Decision:**

Reject

**Comment:**

The authors consider the important problem of developing generalization bounds for stationary mixing processes. This paper uses the recent notion of online-to-PAC conversion, with small modification, together with regret bounds/techniques for online learning with delayed feedback, in order to derive these generalization bounds. This decision is difficult. On the one hand, simplicity of a paper should not be viewed as a bad thing. On the other hand, this work seems premature in that it is not clear how the new generalization error bounds shown here go beyond what was previously known; it feels like the authors simply need to expose a "killer application" to make this work significant.

On the technical side, the present paper simply joins online-to-PAC conversion (i.e., the generalization game) and online learning with delayed feedback. The combination looks straightforward. Various results are presented in the paper, but from my close look I could not identify which results of the authors required technical novelty. The power of the paper comes from previous results on online-to-PAC conversion and previous, well-known-and-understood results from online learning with delayed feedback. As such, the paper feels almost like an application of known results (although, potentially with a nice payoff). Moreover, the authors themselves, in their author response, did not claim technical novelty (multiple reviewers had asked about this).

However, even if new technical tools have not been developed, the paper may still have sufficient merit to be accepted if the results themselves (the new generalization error bounds) are significant. Unfortunately, it is here that the story becomes murky; at this time, for the reasons explained below, I believe the paper is not ready to be accepted.

A first major issue (which was raised in different ways by some reviewers) is the question of how the authors’ Assumption 1 (here, I imagine the case of "geometrically mixing") compares to the assumption of $\beta$-mixing from earlier work; see, e.g., the cited work of Mohri and Rostamizadeh (2008). The authors appear to claim their assumption is weaker in the case of bounded losses since their assumption is at the loss level and one-sided. However, are there actual, non-contrived scenarios of interest where the authors’ assumption truly is weaker? A detailed assumption comparison is in order where, in particular, the authors point out, in some setting of interest, how their new results advance upon what was previously known (here, I mean the new bounds themselves, rather than the proof techniques).

Next, during the discussion period, the authors mentioned that their results improve upon the work of Mohri and Rostamizadeh (2008) in that the authors obtain PAC-Bayesian bounds. However, PAC-Bayesian bounds have previously been shown for beta-mixing processes. See the work of Ralaivola et al. (2010), specifically, Theorem 19. I understand that there are different flavors/forms of PAC-Bayesian bounds, but certainly the authors should be able to engage in a comparison between their bound (Corollaries 3 or 4) and the result of Ralaivola et al. The authors should also see if there are any later works on PAC-Bayesian bounds for non-i.i.d. data.

Finally, the part of the paper that may have the most promise to showcase the authors techniques may well be Section 5, on dynamic hypotheses. Unfortunately, it is unclear how we should view the new results of the authors in light of previous work. There were some brief comments in the Introduction section, which I thought foreshadowed a richer discussion in Section 5. However, Section 5 lacks discussion comparing the authors' results to previous work. In particular, are there applications of interest where the authors results are better than what was previously known? Here, a comparison to the recent works of Eringis et al. and the less-recent works involving Alquier and Wintenberger (/Alquier, Li, Wintenberger) is necessary. For example, for the settings where Eringis et al.'s results also apply, do the authors obtain better results? Next, are there meaningful settings that the authors can handle that cannot be handled by the results of Eringis et al.?
Based on the writing in the Introduction, I feel that the authors know how to do this comparison. Yet, this important comparison was left out of paper.

Overall, I am optimistic that the simplicity/elegance of the authors’ approach should allow them to clearly show a theoretical advance (in terms of their bounds) over previous results. However, adding the detailed discussion with prior results will take some work, and having this context is critical during the review process. Therefore, I cannot argue for this paper to be accepted at this time.

### References

Ralaivola, L., Szafranski, M. and Stempfel, G., 2010. Chromatic PAC-Bayes bounds for non-iid data: Applications to ranking and stationary β-mixing processes. The Journal of Machine Learning Research, 11, pp.1927-1956.